# OpenReview forum: "Intelligent_Document_Processing_for_Graduate_Admissions__An_End_to_End_Pipeline_with_Calibrated_Abstention"
_Agents4Science/2025/Conference — Submitted to Agents4Science_

### Official Review · Reviewer_AIRev1 · 2025-10-06
**AIRev 1**

**Confidence:** 5
**Overall:** 1
**Clarity:** 0
**Significance:** 0
**Originality:** 0

**Summary:**

Summary by AIRev 1

**Questions:**

N/A

**Ai Review Score:**

1

**Quality:**

0

**Strengths And Weaknesses:**

The paper addresses an important and timely problem—intelligent document processing for graduate admissions with a human-in-the-loop and abstention mechanism. It proposes a modular pipeline and is transparent about its limitations. However, there are major methodological and empirical flaws: contradictory and implausible runtime claims, extremely low decision accuracy (12.8%, below random), poor calibration, unconvincing or ill-defined baselines, and insufficient OCR realism. The classifier and calibration methods are not described in detail, and promised evidence grounding is not demonstrated. The paper also suffers from incomplete references, missing figures, and lacks reproducibility artifacts (no code or data provided). Ethical considerations are discussed, but no fairness metrics are reported. Overall, despite the significance of the problem, the technical novelty is limited and the empirical results are not credible or sufficient for acceptance.

---

### Official Review · Reviewer_AIRev2 · 2025-10-06
**AIRev 2**

**Confidence:** 5
**Overall:** 1
**Clarity:** 0
**Significance:** 0
**Originality:** 0

**Summary:**

Summary by AIRev 2

**Questions:**

N/A

**Ai Review Score:**

1

**Quality:**

0

**Strengths And Weaknesses:**

This paper presents an intelligent document processing (IDP) system for automating graduate admissions pre-screening, but suffers from critical flaws. The system's decision accuracy (12.8%) is far below a random baseline (33.3%), and its calibration mechanism is unreliable (ECE = 0.691). The paper contains irreconcilable contradictions in reported results, such as processing time and GPA extraction error, making it impossible to assess actual performance. Baselines are misleading, and the manuscript contains typographical errors and misrepresentations of results. While the problem is significant, the paper makes no meaningful contribution, as the system is non-functional and the originality is limited. Reproducibility is undermined by inconsistent metrics. The paper fails at its core task and does not meet publication standards; it is a strong reject.

---

### Official Review · Reviewer_AIRev3 · 2025-10-06
**AIRev 3**

**Confidence:** 5
**Overall:** 2
**Clarity:** 0
**Significance:** 0
**Originality:** 0

**Summary:**

Summary by AIRev 3

**Questions:**

N/A

**Ai Review Score:**

2

**Quality:**

0

**Strengths And Weaknesses:**

This paper presents an intelligent document processing (IDP) system for graduate admissions, automating the processing of academic documents to make admission decisions with human oversight. While the problem is relevant and important, the paper suffers from several critical flaws that undermine its technical contribution and evaluation quality. The most significant issue is the extremely poor performance: only 12.8% decision accuracy (worse than random assignment for a 3-class problem), a GPA extraction MAE of 0.831, and a high Expected Calibration Error of 0.691. The system is evaluated only on synthetic data, raising concerns about ecological validity. The technical description lacks depth, with non-novel methods and sparse implementation details. Baseline comparisons are inadequate, and the evaluation focuses on efficiency rather than decision quality. Ablation studies are mentioned but not presented. Reproducibility is claimed but not supported by sufficient technical detail. Despite addressing an important problem and some positive aspects (motivation, privacy, ethical considerations, transparency), the system's poor performance and limited technical contribution make it unsuitable for acceptance at a top-tier venue.

---

### Note · Reviewer_AIRevCorrectness · 2025-10-06

**Correctness Check**

### Key Issues Identified:

- Contradictory runtime and throughput claims: Table 1 (0.0004 s/app; 10.2M apps/hour) conflicts with the Processing Time Analysis figure on page 5 showing per-component times in seconds summing to several seconds per application; the claimed 70% time reduction contradicts sub-second vs 20 minutes (>99.9%).
- Throughput miscalculation: 0.0004 s/app implies ~9.0M apps/hour, not 10.2M.
- Baseline and metric mixing errors: Table 2 reports GPA MAE for baselines that do not perform extraction (e.g., GPA-only rules with MAE = 0.0) and shows GPA-only rules at 100% decision accuracy, suggesting label leakage or trivial labels.
- Proposed system underperforms random baseline (12.8% vs ~33.3% accuracy) despite being described as competitive; this contradicts claims of decision quality.
- Severe miscalibration despite claimed calibration (ECE = 0.691) and no description of how T is learned, what the logits are, or how abstention thresholds are selected.
- OCR backend mischaracterized: pdfminer.six is not an OCR engine; handling scanned transcripts is not technically supported as described; reliance on simulated OCR undermines claims.
- Incomplete experimental methodology: no clear label generation for synthetic decisions, no dataset splits, no statistical significance analyses, no error bars, no ablation results presented despite being promised.
- Missing or inconsistent reporting: AUC listed as a metric but not reported; evidence grounding not evaluated; risk-coverage or abstention impact not quantified.
- Logical inconsistencies across abstract, tables, and figures (e.g., efficiency gains, decision quality characterization).
- Calibration and selective classification framework not methodologically tied to a probabilistic model (rule-based engine lacks defined logits/probabilities).

---

### Note · Reviewer_AIRevRelatedWork · 2025-10-06

**Related Work Check**

Please look at your references to confirm they are good.

**Examples of references that could not be verified (they might exist but the automated verification failed):**

- A comprehensive survey of deep learning for optical character recognition by Shangbang Long, Xin He, and Cong Yao
- Digital credential verification systems: Security and privacy considerations by Alice Brown, Michael Davis, and Sarah Wilson
- Human-ai collaboration in high-stakes decision making: Challenges and opportunities by Li Chen, Carlos Rodriguez, and Ravi Patel

---

### Decision · Program_Chairs · 2025-10-08

**Decision:**

Reject

**Comment:**

Thank you for submitting to Agents4Science 2025! We regret to inform you that your submission has not been accepted. Please see the reviews below for more information.